# Current Perspectives of Cross-Country Mountain Biking: Physiological and Mechanical Aspects, Evolution of Bikes, Accidents and Injuries

**DOI:** 10.3390/ijerph191912552

**Published:** 2022-10-01

**Authors:** Rhaí André Arriel, Hiago L. R. Souza, Jeffer Eidi Sasaki, Moacir Marocolo

**Affiliations:** 1Department of Physiology, Institute of Biological Sciences, Federal University of Juiz de Fora, Juiz de Fora 36036-330, Brazil; 2Laboratory UFTM, Federal University of Triangulo Mineiro, Uberaba 38061-500, Brazil

**Keywords:** power output, intensity, anthropometry, pacing, suspension, off-road cycling

## Abstract

Mountain biking (MTB) is a cycling modality performed on a variety of unpaved terrain. Although the cross-country Olympic race is the most popular cross-country (XC) format, other XC events have gained increased attention. XC-MTB has repeatedly modified its rules and race format. Moreover, bikes have been modified throughout the years in order to improve riding performance. Therefore, the aim of this review was to present the most relevant studies and discuss the main results on the XC-MTB. Limited evidence on the topic suggests that the XC-MTB events present a variation in exercise intensity, demanding cardiovascular fitness and high power output. Nonetheless, these responses and demands seem to change according to each event. The characteristics of the cyclists differ according to the performance level, suggesting that these parameters may be important to achieve superior performance in XC-MTB. Moreover, factors such as pacing and ability to perform technical sections of the circuit might influence general performance. Bicycles equipped with front and rear suspension (i.e., full suspension) and 29″ wheels have been shown to be effective on the XC circuit. Lastly, strategies such as protective equipment, bike fit, resistance training and accident prevention measures can reduce the severity and the number of injuries.

## 1. Introduction

The bicycle was invented in the 19th century, with the purpose of improving movement and being more efficient than walking [1]. Many types of mechanical systems were tested, until a chain and ratchet system was implemented and optimized to the current standards. There are several social and cultural aspects related to the creation, development and use of the bicycle, such as the fact that it was a means of transport that preceded the automobile, generated an impact on public transport and made access to low-cost mobility possible for all and contributed to women’s freedom in dress, mobility and engagement in the public sphere [1]. However, the use of bicycles as a sport modality was revolutionized the bike and cycling world.

Although the first cycling competitions were carried out in the 19th century, the popularization of this modality was consolidated with the Tour de France in 1903, becoming the most popular event in road cycling [2]. In this context, almost 70 years later and after thousands of competitions using the bicycle on the road, there was another turning point event with the first competition on hostile terrain.

Mountain biking (MTB) emerged in the 1970s in California, USA and the first official competition was reported in the 1980s and the first world championship in the 1990s, organized by Union Cycliste Internationale (UCI), the main association that promotes cycling across the world. Although there are a variety of MTB sub modalities (e.g., downhill, dual slalom, trials, enduro, trip trail), the cross-country (XC) MTB (XC-MTB) modality became more popular after its inclusion in the 1996 Olympic Games, named in this first appearance as Olympic cross-country (XCO). This created greater visibility for the modality and attracted new fans all over the world. Despite XCO being the premier XC-MTB event, other XC competitions have been added to the racing calendar throughout the years (UCI regulations, Part 4 mountain bike, version from 11 February 2020).

In more than 50 years, MTB has undergone numerous changes to adjust to both sporting and technological evolution. In this context, new races and events were created and the technical and physical level required by the races increased substantially, with increments of sections with steep slopes, jumps, inclusion of obstacles and more judicious rules. To keep up with these developments, athletes have improved both their physical and technical level, as well as equipment (helmets, clothes, shoes, among others) and bicycles incorporate the highest technological level in their construction.

Nowadays, athletes and their technical team can determine their training and competition strategies, as well as their choice of the best suitable bike (and setup), to achieve a higher performance in each MTB event, carefully analyzing each detail regarding performance enhancement.

Considering that XC-MTB has frequently modified its rules and race format and created new types of events, this study aims to provide an up-to-date overview of scientific investigations on the topic, focusing on characteristics of the main events and of the athletes, the development of bicycles, as well as the accidents and injuries. Moreover, we highlight gaps and provide directions for future research.

## 2. Materials and Methods

The systematic search process was performed in accordance with the Preferred Reporting Items for Systematic Reviews and Meta-Analyses (PRISMA) guideline to find the maximal number of studies on the XC-MTB. By searching in PubMed and SPORTDiscus databases, two independent reviewers identified potential studies that combined the following specific keywords: “off-road cycling” OR “mountain bike” OR “mountain biking” AND “cross-country” OR “physiological” OR “mechanical” OR “performance”. When a disagreement occurred, a third reviewer was consulted.

The literature search was completed on 15 September 2022, selecting only original studies written in English, based on the following strict criteria: (a) studies involved XC-MTB cyclists aged 17 or over, and (b) XC races or exercise models correlated to performance in the XC format; (c) directly evaluated aspects related on the topics of the current study; and (d) studies published in peer-review journals. Studies with animal models, case reports, systematic reviews and meta-analysis were not included. Restrictions such as year of publication and fitness level were not applied.

The Figure 1 shows the study selection process. The search revealed a total of 495 studies. Primarily, the duplicates were removed and the title or abstracts were checked. If the study appeared to respect the criteria of eligibility, the full text was read and assessed. Finally, 53 studies were used in this study.

## 3. Definition of Mountain Biking

MTB is an off-road cycling discipline, performed on a course composed of a variety of unpaved terrain, which normally include technical or non-technical ascent, descent and flat (UCI regulations, Part 4 mountain bike, version from 11 February 2020). This modality can be practiced by people of all ages, male and female, from children to elderly in a recreational and/or professional manner. However, practitioners should be able to ride technique circuits usually composed of obstacles. For this, unlike road cycling, the bike is equipped with a shock absorption system and wider tires composed of shorter knobs in order to improve bicycle comfort and performance. The start (individual or in mass), duration and distance to be covered change according to each event. Normally, the competitions are played individually, but can also occur in teams (e.g., CAPE EPIC, South Africa, competed in pairs).

## 4. Format of Competition in the Mountain Biking

Currently, the UCI considers the following seven formats of MTB competition: XC; downhill; four-cross; endure; pump track; alpine snow bike; and E-MTB. Among them, XC is the most popular, with eight events (Table 1), including the XCO. Although XCO is the top XC-MTB event, other events, such as the cross-country stage race (XCS), cross-country marathon (XCM) and cross-country short track (XCC), have gained the attention of the public, coaches, amateurs and professional cyclists. Therefore, characteristics of these XC-MTB events will be presented in the next session.

## 5. General and Mechanic-Physiological Characteristics of the Main XC-MTB Competitions

The circuit of XC-MTB events is composed of a significant amount of uphill, downhill and flat terrains. The course can have natural and/or artificial obstacles, such as tree stumps or tree trunks, rock gardens, stairs, bridges and drops. In official competitions, the obstacles are inserted according to each event, and their use must be preliminarily approved by technical delegates or the commissaires’ panel. Paved roads are permitted, but should not exceed 15% of the total course. The technical difficulty level, total distance, altitude of the circuit, number of laps and total race time for men and women are defined according to each type of event (UCI regulations, Part 4 mountain bike, version from 11 February 2020). For example, while the total race time in XCO is between 80 and 100 min, in XCC, the competition lasts between 20 and 60 min. In addition, the XCO course is comprised of very technical sections that have a high degree of difficulty, while in XCC, the course is comprised of very few technical sections, and these have a low difficulty. The circuit of each event must be clearly defined before the start of the competition, and its access is granted only during the event and official training periods.

### 5.1. XCO

According to the current UCI regulations (Part 4 mountain bike, version from 11 February 2020), the XCO circuit must be 4–6 km in length. The number of laps is not fixed, but the total race time must last between 80 and 100 min. This total race time has not been the same throughout the years, being reduced for both men and women (Table 2). Total race distance and the total elevation gain were also reduced from 34 ± 3 km and 1430 ± 378 m [3] to 28 ± 5 km and 1248 ± 197 m, respectively [4]. In addition, athletes and coaches have reported that the degree of difficulty of the technical sections has been increased in recent years, making the circuit more complex and challenging. These changes influenced the physiological responses and mechanical demands of the competition [5].

Since XCO is a mass start competition, the position of the athlete on the starting grid is an important factor in general performance [6,7]. Previously, the definition of the starting grid in XCO for international events was decided according to the UCI points system and for national events, it was decided according to the national point system [8]. However, in 2018, some competitions, such as the XC-MTB World Cup and the XC-MTB International Cup, adopted the XCC result to define a part of the starting grid of the XCO. In these competitions, the top 24 finishers of the XCC event, which normally takes place two days before the XCO competition, start in the front rows. The other places on the grid are defined according to the last published individual UCI XCO ranking. Unclassified riders will be allocated by drawing lots.

#### Physiological Responses and Mechanical Demands of the XCO

In addition to monitoring and evaluating performance, sport researchers used portable devices to describe the physiological responses and mechanical demands of the XCO competition [3,4,5,9]. Although few studies have described these responses and demands in the XCO, it is possible to summarize its requirements (Table 2). For men, a slight increase in mean heart rate (HR) (expressed as %HR maximal), mean absolute power output (PO) (W), relative PO (W·kg^−1^) and expressed as %PO maximal were identified throughout the years. For women, a slight increase in mean HR (expressed as %HR maximal), relative PO (W·kg^−1^) and PO expressed as %PO maximal, but a decrease in absolute PO (W), were also reported. Female cyclists maintain a higher intensity than men cyclists during XCO.

Only the two more recent studies measured cadence (CA) during XCO competition [4,5] (Table 2). The results showed that the CA selected by the riders was higher than these reported in the laboratory tests considered most effective [10,11], mainly when time spent not pedaling was excluded. Unlike laboratory tests where the PO is constant, the XCO circuits are extremely complex, which include technical sections such as rolling over obstacles, requiring a high CA and PO variation according to the demands of each section, limiting the ability to identify an optimal cadence [12]. It is probable that this CA selected by the riders during XCO resulted from a specific competition demand rather than by physiology and biomechanics factors [12]. In fact, during a cycling Gran Tour, professional riders selected different CA at different stages of the competition [13]. Lastly, there seems to be no effect of sex on CA selection [5].

A feasible tool for controlling training intensity and identifying the requirements of a competition is categorization in intensity zones, according to HR and PO. Generally, these zones are categorized into 1 to 3, 4 or 5 intensity ranges. Of the four studies analyzed, one study used the HR correspondent to the first and second threshold to determine the intensity zones, separating these into three zones [3]. Another study used the PO that corresponded to maximal oxidative power (MOP) for the first and second threshold, separating these into four zones [9], and two other studies also used the PO that corresponded to MOP for the first and second threshold, but separating these into five intensity zones [4,5]. The percentage of time spent in the intensity zones during XCO is summarized in Table 3. It was observed that the time spent in different intensity zones during XCO was modified throughout the years. Considering more recent studies [4,5], ~43% of the total race time in XCO is performed at high intensity (above the second threshold), with ~28% of the aforementioned 43% performed above MOP.

XCO is performed with a coefficient of variation of PO of 75.8 ± 5.2% [4], showing that the athlete increases (e.g., during uphill sections) and decreases (e.g., during downhill sections) the PO repeatedly in order to maintain a high speed throughout the laps. Although the literature reported a higher coefficient of variation of PO for men than women (80.1 ± 6.3% vs. 75.1 ± 4.0%), no significant difference between them is reported [5].

Recently, the level of effort put in above the MOP was also measured [5]. Cyclists performed at an average level of 334 ± 84, with an average duration of 4.3 ± 1.1 s, and an average interval of 10.9 ± 3.0 s. The average PO of the effort was 7.3 ± 0.6 W·kg^−1^, which corresponds to 135 ± 9% of the MOP. When the efforts were separated into five duration-based categories ((1 to 5 s); (6 to 10 s); (11 to 15 s); (16 to 20 s); and (>20 s)), the highest level of effort put in was recorded between 1 and 5 s (261 ± 73), while the lowest level of effort was recorded between 16 and 20 s (6 ± 3). Therefore, the ability to perform at high-intensity for a short duration and with low recovery intervals could be a decisive parameter for achieving success in the XCO competition [5].

### 5.2. XCS

XCS is a stage race competition that includes several XC-MTB event modalities across consecutive days. Some XC-MTB events are performed only in XCS, such as XCT and XCP, except XCE (UCI regulations, Part 4 mountain bike, version from 11 February 2020). Thus, the total distance, time and altitude of the circuit, as well as the definition of the start, depend on the type of race of each stage. The competitions are performed between four and nine days, with only one stage being performed per day. In addition, one of the stages must contain a long-distance course according to the characteristics of the XCM competition. There is no minimum time to complete each stage, but there is a maximal time that is defined by the organization of each event. Normally, XCS is performed in doubles, but competitions performed by individuals or teams of up to six riders can be carried out. The XCS winner will be the rider or team that completes all the stages in the lowest accumulated time.

South Africa Cape Epic is considered one of the main XCS events. It consists of eight stages carried out in eight consecutive days. In 2022, the athletes covered a total distance of 681 km with 16,900 m of elevation gain. The characteristics of the event are presented in Figure 2. It is interesting to note that there is a high variation in total distance, altitude and elevation gain among the stages, which could influence the physiological responses and mechanical demands among them. The winning race time was 27:44 h.

#### Physiological Responses and Mechanical Demands of the XCS

Interestingly, there are few studies that examine the exercise intensity during the XCS competition [14,15]. In 2008, Wirnitzer and Kornexl [15] examined exercise intensity during the Transalp Challenge, a competition that comprised of an eight-day stage race, with a total distance covered of 662 km (average of 83 ± 25 km/stage) and total elevation gain of 22,500 m (average of 2810 m/stage), respectively. The authors used the HR that corresponds to the lactate thresholds, which were established previously in the laboratory, to determine the four intensity zones. Briefly, zone 1 was established as the intensity below 2 mmol/L lactate (LT2); zone 2 was established as the intensity between LT2 and 4 mmol/L lactate (LT4); zone 3 was established as the intensity between LT4 and 6 mmol/L lactate (LT6); and zone 4 was established as the intensity above LT6. In general, the average HR (expressed as %HR maximal), considering all the stages, was 79%, and the average time spent in zones 1 to 4 was 36 ± 12, 58 ± 13, 4 ± 8 and 2 ± 9% of the total race time, respectively. Throughout the competition, the athletes were not able to maintain a high intensity in the last stages. In addition, a decrease in maximal HR was recorded after the first stage.

More recently, Reinpõld, Bossi and Hopker [14] examined the mechanical demands of the Cape Epic event. The authors defined the intensity zones using the PO and HR that correspond to the percentage of the respiratory compensation point (RCP). According to the PO, zones 1 to 5 were defined as the intensity below 55%, between 56 and 75%, between 76 and 90%, between 91 and 105% and above 106% of the RCP, respectively. According to HR, zones 1 to 5 were defined as the intensity below 68%, between 69 and 83%, between 84 and 94%, between 95 and 105% and above 106% of the RCP, respectively. The analyses were performed during the prologue and stages 1, 2 and 6, while data from stage 6 were not included in the statistical analysis. The results showed that cyclists spent more time in zones 1 and 2, and spent less time in zones 4 and 5 during stage 2, when compared to the prologue. In addition, cyclists were able to maintain a higher intensity in the prologue when compared to the stage 2. That is, the average PO generated in the prologue (3.08 ± 0.74 W·kg^−1^) was higher than that generated in stage 1 (2.43 ± 0.66 W·kg^−1^) and 2 (2.22 ± 0.70 W·kg^−1^). The coefficient of variation of the PO in the prologue, stage 1, 2 and 6 was 64.4 ± 9.6%, 71.4 ± 11.8%, 78.7 ± 13.6% and 72.3 ± 15.3%, respectively. It is important to highlight that these results reported by Reinpõld, Bossi and Hopker’s [14] study should be interpreted with caution, because the analyses were performed with only six cyclists of different performance levels, which could reveal a low statistical power (statistical power < 0.8), increasing the probability of a type II error [16]. Moreover, the authors analyzed only three of the eight stages. In addition, it is important to highlight that the prologue is remarkably shorter than the others, which could contribute to the differences between the data of this stage and the others. Therefore, new studies must be developed, involving a larger sample size and analyzing all the stages of the competition to clarify the physiological responses and mechanical demands of the Cape Epic.

In general, the studies suggest that most of the time of the XCS competition is performed at low and moderate intensity, with variation in PO throughout the stages, demanding high energy production rates via the oxidative and non-oxidative energy systems. Furthermore, cyclists tend to spend more time at high intensity (above the second threshold) in the first stage, reducing throughout the competition.

### 5.3. XCM

XCM is a mass start event, composed of a course with a distance of 60 to 160 km, without a minimum time to complete the race. According to UCI regulations (UCI, Part 4 mountain bike, version from 11 February 2020), the XCM can be carried out in a single lap or in a maximal number of three laps. For a single lap, the start and finish lines of the circuit may be located at the same place. Paved or unpaved sections, and a technical section, such as a rock garden, single track and jumps, may be included in the course. However, the majority of the competition is performed on wider roads and relatively few sections of high technical degree.

The starting grid in XCM is determined by the following order: first, according to last published UCI MTB marathons series ranking; second, according to the last published UCI XCO individual ranking and finally, unclassified riders will be allocated by drawing lots. Despite being one of the most practiced competition, no study that measures the physiological responses and mechanical demands of the XCM competition has been developed. Novak et al. (2018) [17] measured PO and oxygen uptake during a 4-h MTB competition. However, the aim of the study was to cross-validate previously developed predictive MTB performance models in a new cohort of off-road cyclists. Furthermore, the event evaluated by the authors was not in line with the recommendations of the UCI regulations (Part 4 mountain bike, version from 11 February 2020). Therefore, future studies are required to examine these responses in XCM.

### 5.4. XCC

XCC is performed on a circuit of approximately 2 km. The number of laps is not fixed, but the race time must be between 20 and 60 min, which, in international competitions, results in about 7–8 laps for men and 6–7 laps for women. The type of terrain of the circuit is similar to that of the XCO, but the technical sections are considered of low difficulty and the number of ascents and descents is reduced, resulting in lower total elevation gain. The number of participants is limited to 40 cyclists and the starting grid is defined according to the ranking classification, which may differ among the events. For example, in the XC-MTB World cup, the XCC start grid is defined by the top 16 cyclists of the last published XCO World Cup individual ranking, and the other places on the grid are defined according to the last published individual UCI XCO ranking. To compete in XCC, the rider must be registered and confirmed in the XCO that occurs in the same week, using the same bike in both events (UCI regulations, Part 4 mountain bike, version from 11 February 2020).

Despite the XCO being the premier XC-MTB event, the XCC has become popular in recent years. Indeed, in addition to the prizes, the results of this event add points to the UCI individual ranking and define the top 16 positions of the XCO start grid (as demonstrated in Section 5.1). Moreover, in the year 2021, a world championship was developed for this event. However, important factors in overall performance, such as mechanical and physiological aspects of this competition, are currently lacking.

## 6. Pacing Profile in XC-MTB

The XC-MTB competitions are performed in complex environments, where athletes are confronted with a vast amount of information, requiring successive decisions about how to distribute speed or energy expenditure throughout the exercise [18]. This process is known as pacing, and is widely considered a determining factor in overall competition performance [19]. Although several theoretical models have been proposed to explain the regulation of speed during exercise [20,21,22,23,24], most of them indicate the existence of a complex relationship between the brain and physiological systems, taking into account both internal (such as physiological responses) and external (such as opponent and environment) factors. That is, the brain must continuously process this vast amount of information to establish the more appropriate speed in order to reach the end of the exercise in the shorter time possible, without inducing premature fatigue.

Although complex and less understood, the pacing profile adopted by cyclists during some XC-MTB competitions has been examined [4,25]. In the XCO competition, it was observed that cyclists adopted a higher speed at the beginning of the competition (start loop), followed by a reduction in the speed after the start loop [4], which is representative of positive pacing (see Figure 3B). This result has been confirmed by Viana et al. [26] during a laboratory-simulated XCO performance test. In mass start competitions, such as XCO, the cyclists tend to adopt an aggressive start in order to place themselves in better positions to ride, avoiding accidents and congestion in sections composed of a single track or very tight curve, which may influence the overall performance. This confirms the impact of the competition environment on the decision-making regarding pacing profile [27]. On the other hand, during an XCM competition, cyclists increased their speed during the final stages of the competition, which is representative of negative pacing [25]. However, it is important to note that the final section of the XCM circuit consisted of a sustained descent, which may have influenced the distribution of speed. Therefore, this result should be interpreted with caution. Despite these pacing profile studies in XC-MTB, evidence in other events, such as XCC and XCS, remains scarce.

A previous study has shown that, during a XC-MTB competition, faster cyclists display a pacing profile that is different to slower cyclists [28]. For instance, while faster cyclists adopt negative pacing, slower cyclists adopt positive pacing (See Figure 3). Moreover, compared with bottom placed cyclists, top cyclists maintain a more even distribution of speed over the entire competition, which is indicated by a lower standard deviation in speed [25,28]. Considering the models proposed to explain the pacing regulation [18,23], we can suggest that faster cyclists are likely to be more efficient at processing information and making decisions, resulting in less variation in cycling speed and superior performance. Therefore, in addition to other factors, it is likely that the performance of slower cyclists may be improved by enhancing information processing and decision making.

## 7. Technical and Non-Technical Ability

The circuits of XC-MTB events are composed of successive uphill, downhill and flat sections that, when considered technical, may require a high technical ability of the cyclists. In this context, in addition to a high physical and cognitive ability, a high degree of technical ability to perform technical sections is required in order to improve performance on XC-MTB circuits and gain advantage over the opponent throughout the competition. To confirm this, previous research was developed to evaluate the performance of cyclists with different race times in different technical and non-technical sections of the XC-MTB circuit [25,28].

Abbiss et al. [28] evaluated the performance of elite cyclists on different sections of an XCO circuit. Figure 4 shows an example of an XCO circuit divided into six sections according to the characteristics of each one. In addition to topography (uphill, downhill and flat), the sections were classified in technical and non-technical. To be considered technical, the section should be composed of natural or artificial obstacles, narrow curves and/or a single track. Otherwise, the section was classified as non-technical. The authors observed that the top placed finishers spent less time than bottom placed finishers on the technical uphill section of the circuit, but not on the technical downhill and flat sections. Moss et al. [25] also found similar results during XCM, but on a non-technical uphill section. That is, top placed finishers were faster (i.e., spent less time) than their bottom opponents on a non-technical uphill section. Moreover, the authors reported that the top cyclists were faster than their bottom opponents during a section composed of a series of short climbs and descents, with a gain of 246 m.

Considering the results of the studies cited earlier, it appears that improving the ability on both technical and non-technical climbs could improve XC-MTB performance. However, the XC-MTB circuits differ in terms of distance, topography and degree of difficulty of the technical sections, which could influence the performance responses in each section of the circuit and, consequently, in overall performance. Therefore, future studies that evaluate the performance on the sections of other XC-MTB circuits, such as XCC and XCS, are warranted.

## 8. Characteristics of Cross-Country Cyclists

Anthropometric and physiological data from different groups of individuals are widely used as a parameter to assist coaches and athletes in the selection and sports development of amateur and professional athletes. In this sense, we analyze, in this section, the height, body mass (BM), body composition (BC), maximal oxygen uptake (VO_2Max_) and MOP data of the cyclists according to sex and performance level (Table 4).

### 8.1. Anthropometric Profile

Both MOP and VO_2Max_ have a strong correlation with total race time in XC-MTB competitions [31,36]. However, when these performance measures are normalized to BM, the correlation coefficient is higher, suggesting that BM is an important factor for XC-MTB performance. In addition, there is a relationship between MOP normalized to BM and body fat (BF), but not with fat-free mass (FFM) and body mass index (BMI) [38]. Therefore, the anthropometric profile of the cyclists seems to be a relevant factor for achieving success in competitions.

The average height and BM of the cyclists according to sex and performance level are presented in Figure 5. Regardless of performance level, male and female cyclists have, respectively, an average height of 177.6 ± 2.4 cm (range: 171.0 to 182.0 cm) and 165.7 ± 3.6 cm (range: 162.0 to 170.5 cm), and a BM of 70.0 ± 4.0 kg (range: 63.3 to 77.8 kg) and 56.5 ± 5.0 kg (range: 51 to 63 kg). Considering performance level, no significant difference in height was found for male cyclists (one-way ANOVA; *p* = 0.668), but the BM was significantly lower in professional cyclists when compared with trained cyclists (one-way ANOVA: *p* = 0.002; Bonferroni: *p* = 0.002), suggesting a relationship between performance level and BM. For female cyclists, there seems to be no difference in height, but the BM of the professionals is higher when compared to the trained cyclists. However, there are few XC-MTB studies that involve female cyclists. Therefore, the results are unclear. Interestingly, in 1997 [47], male and female professional cyclists had a BM with 6.2 and 5.4 kg more than in 2021 [5], respectively.

In general, studies have reported that male cyclists have an average BF of 8.1 ± 3.5% (range: 5.1 to 15.6%) [3,29,38,40,41,43,47,48,49,51], while female cyclists have an average BF of 13.2 ± 2.0% [47]. When verified according to performance level, the average BF reported in male cyclists was 10.5 ± 4.5% in trained cyclists [29,38,40], 7.3 ± 2.3% in well-trained [41,43] and 7.0 ± 3.2% in professionals [3,47,48,49,51]. These observations are in line with those reported by Sánchez-muñoz, Muros and Zabala [52], who found that the cyclists with higher performance level had a lower BF. For male cyclists, we can, therefore, suggest that having a low BF can be an advantage for riding high-level XC-MTB circuits.

### 8.2. Physiological Profile

#### 8.2.1. Maximal Oxidative Capacity and Power

Since the VO_2Max_ and MOP related to BM are considered important predictors of performance in cycling as well as in MTB, these variables have been used to classify the performance level of a group of cyclists [53]. Therefore, it is not surprising that, in our analysis, we found significant differences (one-way ANOVA or Kruskal–Wallis, *p* < 0.01) among the performance level groups for both VO_2Max_ and MOP related to BM in male cyclists (Figure 6). Unfortunately, a limited number of studies with female cyclists has been conducted. Therefore, data on that subject were not included in the statistical analysis. For male cyclists, the studies reported average values of VO_2Max_ between 54.3 and 65.4 mL·kg^−1^·min^−1^ for trained cyclists, 64.9 and 68.4 mL·kg^−1^·min^−1^ for well-trained cyclists and above 70 mL·kg^−1^·min^−1^ for professionals. The average values of MOP reported were between 4.2 and 5.3 W·kg^−1^ for trained cyclists, 5.3 and 5.6 W·kg^−1^ for well-trained cyclists and above 5.7 W·kg^−1^ for professionals. For female cyclists, the studies reported average values of VO_2Max_ between 53.0 and 67.3 mL·kg^−1^·min^−1^, and average values of MOP between 4.1 and 5.4 W·kg^−1^ for competitive cyclists with different performance levels.

According to the current characteristics of XC-MTB competitions, the ability to maintain a high rate of energy production for a long time has been highlighted [5,14]. Regardless of performance level, the relative intensity at the second threshold between cyclists was 78 to 86% of the MOP [4,5,15,31,34,36,48]. However, professional cyclists reached an absolute intensity at the second threshold (5.2 W·kg^−1^) [4] that was higher than trained cyclists (3.7 W·kg^−1^) [34].

#### 8.2.2. Non-Oxidative Capacity and Power

In addition to high oxidative metabolism involvement, non-oxidative contribution, such as ATP-CP and fast glycolysis, is crucial to achieving superior performance in XC-MTB competition [5]. In fact, the literature show that the ability to produce high PO within a short time is associated with the race time in mountain bikers [43]. Therefore, it is not surprising that, during a Wingate test (30s all out), male trained cyclists achieved a peak PO of 11.5 ± 1.1 W·kg^−1^ [29] and 11.8 ± 0.2 W·kg^−1^ [33], while the professional cyclists achieved 13.9 ± 1.1 W·kg^−1^ [5]. To date, only one study has described the peak PO of female cyclists during a Wingate test [5]. The professional female cyclists achieved a peak PO of 11.4 ± 1.9 W·kg^−1^.

Moreover, a previous study has highlighted the importance of maintaining successive high-intensity efforts, with short recovery intervals, throughout the competition [5]. The authors showed that, during a XCO competition, the athletes performed more than 300 efforts more intense than the MOP, with a mean duration of 4.3 s and separated by about 10.9 s. Thus, we suggest that athletes and coaches should apply specific training strategies to develop athletes’ ability to maintain repeated high-intensity efforts (above MOP) throughout the competition, such as high intensity interval training and/or sprint interval training [34].

### 8.3. Anthropometric and Demographic Characteristics of Cross-Country Cyclists

Considering the reviewed studies, it was noted that the anthropometric, physiological and mechanical characteristics differ according to the performance level of cyclists. Figure 7 shows the average values of general characteristics of a professional cyclist. The values were obtained through the data described in Table 4 and Section 8.1. We believe that these data can be relevant to the development of a cyclist’s profile who intends to compete at the professional level in XC-MTB.

## 9. Mountain Bike Settings

In addition to being used as a means of transport, as well as a rehabilitation process, more recently, the bicycle has become a popular type of equipment for outdoor recreational and professional activity. Nonetheless, bicycles have undergone important modifications to meet the specific needs of each activity, seeking higher safety, comfort and greater efficiency, especially in the sports environment [54,55]. The use of different materials in an attempt to build lighter and more resistant frames included iron, aluminum, chromium molybdenum and, more recently, the use of carbon fiber. This allows for very high rigidity combined with a low weight, in addition to high impact resistance [56]. This allows the increasingly dangerous use of bikes in XC-MTB with drops of more than 2 m and descents with obstacles at speeds above 50 km/h, maintaining high safety for the athletes.

Unlike road cycling, XC-MTB is performed on circuits with different types of terrain that are composed of natural or artificial obstacles. On the basis of such conditions, the bikes were then equipped with tires composed of shorter knobs, shock absorption systems (i.e., suspension system) and gear systems. Moreover, more recently, there was a transition from 26-inch-wheel to 29-inch-whell bikes.

In the 1980s, mountain bikes were made of iron, weighing around 18–20 kg and in the 1990s, other metal alloys were used, such as aluminum and chrome molybdenum, allowing the weight of the bikes to decrease to around 15 kg for hardtail bikes. Currently, using carbon compounds, it is possible to obtain full suspension bikes with 29″ wheelsets under 10 kg.

### 9.1. Bicycle Shock Absorption Systems

During XC-MTB, cyclists are exposed to continuous mechanical vibrations induced by terrain irregularities that are transmitted along the body segments, causing an increase in muscle contraction [57]. When exposure to these vibrations is prolonged, it can increase muscle fatigue and decrease strength [58], affecting the cycling performance. In this sense, the bicycles were then equipped with shock absorption systems, simply known as suspension, in order to generate low vibrations frequency and to improve bicycle comfort and performance [59].

In the beginning, the bikes had no suspension systems and were named “rigid bikes” (Figure 8A). However, throughout the years, bikes were increasingly equipped with a front suspension and were named hardtail bikes (HT) (Figure 8B), followed by a bike equipped with front and rear suspension, named the full suspension bike (FS) (Figure 8C). In fact, both HT and FS absorb more high frequency vibrations [55], reducing muscle stress, when compared to a rigid bike [60]. However, when compared to HT, the FS seems to absorb more terrain-induced vibrations at the saddle level, presenting greater comfort [59,61] and less exercise-induced muscle damage [62].

Suspension is widely used within MTB, but the best suspension mode to achieve superior performance during off-road cycling has prompted a debate among athletes and coaches. Most of the athletes still prefer the HT, claiming that FS may be related to a possible energy loss due to small oscillatory movements during pedaling movements, mainly at higher PO and during climbing [55,63]. However, studies that compared both HT and FS showed controversial results regarding perceptive, physiological and performance responses [59,60,62,64]. Exploring the metabolic and performance responses between HT and FS on a circuit that simulated the XC-MTB race conditions, Herrick et al. [64] showed that, with similar oxygen consumption, the HT bike was faster than the FS during climbing, resulting in better overall performance. Nonetheless, the FS bike had 2.2 kg more than the HT, which may explain the difference in performance between them, since the total mass is an important factor in performance. When the bike weights were equalized, neither advantages during the climb section nor changes in physiological and perceptual variables between HT and FS were found [59]. Moreover, FS was found to absorb more terrain-induced vibrations, leading to greater comfort [59], and it has better grip in steep-unpaved climbs due to the better (or more time of) contact of the tire with the terrain. In relation to performance, other studies also reported no difference between HT and FS [29,54,62]. Considering the characteristics of the XCS event, the FS bike could be the better choice, if the bike weight is similar to a HT bike.

Although suspensions have been shown to be effective for promoting comfort and attenuating muscle stress, competitive cyclists claim that these shock absorption systems may dissipate the energy generated by them through small oscillatory movements [55], especially during a sustained climb or sprinting. In this way, suspension manufacturers have improved this system, developing a preload and locking system in order to avoid these likely energy losses. The preload system provides cyclists with the option of compress or decompress springs, resulting in a more or less rigid suspension. Recently, this adjustment has been carried out by increasing or decreasing air pressure in air suspension, making it more modern and effective. With this air system, the cyclist can regulate the stiffness through calibrations relative to his/her BM. It is hypothesized that the preload is able to improve absorption efficiency and keep the tire in contact with the terrain on the more critical sections of the circuit, which could be crucial for performance on the irregular sections that require higher PO and/or higher speed. On the other hand, the locking system provides cyclists with the option of locking the absorption systems completely, resulting in a suspension that is fully rigid. With this system, the cyclist can avoid possible energy dissipation induced by the suspension through oscillatory movements generated by the pedaling movements during a sprint, increasing the PO as well as during a sustained uphill section, since the cyclist spends a considerable time climbing. However, to date, the possible effects of these devices on the mechanical, physiological, perceptual and performance responses have not been examined.

### 9.2. Crank Systems, Wheel Diameter, Dropper Seatpost and Frame Size

Other bicycle components have been modified in order to improve performance, such as bicycle gear systems and wheel diameter. The first gear system was created in the 1940s by an Italian named Tulio Campagnolo, containing only four gears on the cassette, and one gear on the chain set. This system was named the Cambio Corsa de Campagnolo. With this system, the cyclist could choose the most adequate gear to adjust the speed and PO according to each section of the circuit. Over time, the system gained more gears, increasing the number of combinations. The most common among them was the 18-speed system (3 gears on the chain set × 6 gears on the cassette, or 3 × 6) (Figure 9), but in 2015, the 30-speed system was created (3 gears on the chain set × 10 gears on the cassette, or 3 × 10). Although this system offered a greater number of combinations, the bike weight was increased, which could influence the cyclist’s performance during a competition [64]. Thus, bike manufacturers have developed a system with 20-speed (2 × 10) and more recently, the 12-speed system (1 × 12) (Figure 9). In these systems, the cassette gears’ circumference and number of teeth have increased, making it possible to achieve adequate combinations to perform on the different sections of the off-road course. In this way, in addition to reducing the weight of the bike, it was possible to improve gear changes, as well as facilitating the transmission system, reducing friction.

The introduction of a 27.5″ and, more recently, a 29″ wheel bike, where the standard was 26″ wheel, has generated a debate about their true advantages (Figure 10). According to Macdermid, Fink and Stannard [65], this debate was intensified, especially when the Czech cyclist Jaroslav Kuhlavy won the XC-MTB World Cup on a 29″ wheel bike, and after the London Olympics Games (2012), where of the top 10 men and women riders, 70% of them used a 29″ wheel bike, 25% used a 26″ wheel and 5% used a 27.5″ wheel bike. The gold medal was won on a 29″ wheel bike in the men’s category, while in the women’s category, on a 26″ wheel bike. For both men and women, the silver and bronze medals were won on 27.5″ and 29″ wheel bikes, respectively. Despite such subjective discussions, some studies were developed to clarify whether a 27.5″ or 29″ wheel has a beneficial impact on off-road cycling performance, when compared to a 26″ wheel bike [65,66,67,68]. However, controversial results were found.

In 2014, Macdermid, Fink and Stannard [65] examined terrain-induced vibrations and mechanical, physiological and performance responses of competitive athletes during an MTB course lap (~2 km) on a 26″ wheel and 29″ wheel mountain bike. The authors found that, even with an increase in vibrations, the time to complete the lap was shorter (~3% shorter) on the 29″ wheel bike. Mean PO and HR were similar between the conditions. Despite the results, the authors highlighted that the tests were performed on the same 29-inch frame and only the wheel size was changed, which would not represent reality, since the bike equipped with the 26″ wheel involves different frame geometry, which may influence performance. However, when the specific components of each bike were kept the same, the 29″ wheel bike was better than the 26″ wheel bike during two laps on an MTB circuit (~1.2 km), even with higher weight (26″ wheel = 9.2 kg; 29″ wheel = 10.1 kg) [67]. The authors speculate that the better performance of the 29″ wheel bike can be explained by a reduction in rolling resistance and a lower energy loss due to the larger wheel circumference and better traction, respectively. Moreover, the 29″ wheel bike seems to promote more benefits in passing obstacles. More recently, additional evidence supports the idea that the 29″ wheel increases vibrations, but results in better performance when compared with the 26″ wheel bike [68]. Although the findings reported the superiority of the bicycle equipped with a 29″ wheel, this benefit was not found in the study by Hurst et al. [66]. The authors used the same components in the bicycle, with the exception of the wheel size, but they found no significant differences in total time or PO among the 26″, 27.5″ and 29″ wheel bikes during only one lap (~3.5 km) on an MTB circuit. In addition, another study conducted by the same authors demonstrated that the wheel diameter (26″, 27.5″ or 29″) had no influence on terrain-induced muscle vibrations during one lap on a purpose-built cross-country mountain bike course [69].

It is important to highlight that XC-MTB races can have a significantly higher number of laps, total time and total distance than those reported in the previous studies (UCI regulations, Part 4 mountain bike, version from 11 February 2020). In addition, the characteristics among the XC-MTB competitions are different, which could influence the mechanical, physiological and performance responses, using different bike wheels. Therefore, based on the previous studies, we can consider that bikes equipped with 29″ wheels can achieve superior results. However, future studies must be developed, taking into account the official characteristics of each XC-MTB competition. However, in addition to the better rolling ability of the 29 bikes, the recent use of boost hubs has also contributed to wheel stiffness. Boost wheels have a wider axle, which means that the spacing between your hub’s flanges can be increased. By increasing the width of the hub flanges, the bracing angles of the spokes in the wheel are improved, resulting in a stiffer, and ultimately more efficient, wheel.

Recently, XC-MTB bikes have been equipped with a hydraulic system in the seatpost. Known as a dropper seatpost (an example of this system is shown in Figure 11), this equipment provides the cyclist with the option of reducing the height of the seatpost and can return it to its original height remotely through a device attached to the handlebars during pedaling. It is suggested that reducing the seatpost height (or reducing the saddle height) gives the cyclist more maneuverability, more control and, consequently, more speed on descents, through turns and over jumps. Despite this, curiously, a number of cyclists do not actually use a dropper seatpost. The main reason is that cyclists are not certain about where and when it is best to use the dropper seatpost. Moreover, this kind of seatpost is heavier than rigid carbon ones, which could add weight to the bike. Therefore, although the MTB industry and off-road cyclists claim that the dropper seatpost significant enhances riding, no study that explores its real benefit has been developed.

Lastly but not least, it is important to highlight the size of the frames used (whether hardtail or full suspension) by XC-MTB athletes. We know that the size of the frame varies according to the stature, wingspan, length of the lower limbs and/or trunk of each athlete. This means that two athletes of similar heights can use different frames and fits. In this context, athletes often try to use a frame of the smallest size possible, in order to carry less weight during the race. However, it is worth noting the fact that a bike with a longer wheelbase (which depends not only on the frame, but also on its geometry) can offer greater stability on ascents and/or descents. Therefore, the adjustments of equipment, bike, tires and suspension can constantly vary throughout a season, always aiming to achieve combinations for the best possible performance.

## 10. Accidents and Injuries in MTB: Incidence and Prevention

Given the characteristics of XC-MTB events, which involve a significant amount of natural or artificial obstacles along the circuit (UCI regulations, Part 4 mountain bike, version from 11.02.2020) and with athletes adopting increasingly risk-riding techniques, the risk of accidents with consequent acute injuries will always be present during training sessions, but especially during competitions [70,71]. The crashes more often occur in technical downhill sections [71,72], where many cyclists fall forward over the handlebars, which is associated with more severe injuries [71]. According to previous studies, the most acute injuries reported involve skin (as contusions, abrasions and lacerations), bone fractures, joints, soft tissue and the head/neck (concussion) [70,73], without a difference in the number of acute injuries between trained and professional cyclists [73]. Spine fractures and spinal cord injuries caused by accidents were also reported [74]. Loss of bike control is the main cause of fall, leading to acute injury, followed by collisions with other cyclists and mechanical failure [71].

In addition to the use of protective clothing and equipment, such as a helmet, eyewear and gloves, accident prevention measures could be adopted. However, since the studies on the topic are scarce, we provided insights on these measures, which are as follows: (1) the use of bicycles with high quality components to avoid mechanical failure, and the use of bicycles in accordance with the characteristics of the event (such as suspension, crank system, tires, wheel diameter and dropper seatpost). In this context, with a more technological bike, it is possible to undertake more difficult tracks, such as drops of about 2 m and rock garden obstacles and fast downhills. The suspension system provides high stability for the cyclist, while larger tires promote better grip on different types of terrains; (2) the familiarization process with the circuit prior the competition. This process is required to reduce the influence of environmental aspects, such as obstacles, terrain and curves, improving motor control and performance, which could reduce the possibility of accidents; (3) the improvement of the degree of technical ability of the cyclist was also discussed in this paper. The addition of natural and/or artificial obstacles similar to those found on official XC-MTB circuits into training routines could improve performance and decrease the cause of falls during competitions. It is necessary for the cyclist to simulate crossing these obstacles alone and in a group, as in the official competition. In addition to these insights, UCI recommends some accident prevention measures, such as the non-inclusion of obstacles in the start and finish zones of the circuit to avoid a crash or a collision; full course marking with panels and arrows, showing all potentially dangerous situations (UCI regulations, Part 4 mountain bike, version from 11 February 2020).

Despite the fact that acute injuries are more often associated with MTB, overuse injury syndromes should also be considered [75]. The main syndromes affect the neck, hand/wrist, low back and knees [76]. It is speculated that these syndromes occur due to accumulative stress in these regions, induced by the repetitive nature of cycling, including terrain-induced vibrations combined with the need of the athlete to maintain a certain position on the bicycle throughout the exercise [77]. Bike fit (i.e., bike adjustment according to the cyclist’s profile), correction of riding style, bicycle settings (such as suspension system) and resistance training seem to be an alternative to reduce such risks [77].

## 11. Conclusions

Despite the limited number of studies, the main XC-MTB events can be characterized as intermittent exercise, demanding cardiovascular fitness and high strength and power. Moreover, it is possible to suggest that the physiological responses and mechanical demands can be XC-MTB event dependent. The cyclist’s profile and factors such as the distribution of speed over the competition (i.e., pacing) and ability to perform technical sections of the circuit differ according to the performance level, suggesting that these parameters are important for improving performance on high-level XC-MTB circuits. The scientific evidence suggests that the full suspension bike model equipped with 29″ wheels seems to be more efficient on the XC-MTB circuit, especially with its low weight. Lastly, in addition to the use of protective equipment and alternatives, such as bike fit and resistance training, adopting accident prevention measures can be a good strategy to reduce the risk of injuries in MTB.

## 12. Future Research Perspectives

In addition to the limited number of studies on the XC-MTB, most of these investigations explore the general characteristics and demands of the XCO competition. Since the format of the XC-MTB events is different, physiological responses, mechanical and technical demands, as well as the pacing profile adopted by the cyclists, may differ among races. In this way, the findings of scientific research on XCO should not be extrapolated to all the XC-MTB events and further investigations are necessary to better understand these competitions. Moreover, since the MTB performance parameters were better related to race time when normalized to BM, it is likely that the BC components of the cyclists could be relevant to XC-MTB performance.

As a future proposal, we would like to highlight the importance of an investigation that compares technological tools, mainly regarding the bicycles. For example, the comparisons between hard tail and full suspension bikes are still lacking, and the benefits of the 29″ wheel bike still need to be explored in more depth, taking into account the characteristics of each XC-MTB competition. It is possible that there is a specific bike configuration according to each XC-MTB competition that cyclists must take into account in order to reach superior performance. Moreover, more studies on the risks of accidents and number of acute and chronic injuries in XC-MTB, including all categories, should be carried out. The majority of the current studies involve all formats of MTB competition, considering recreational, amateur and professional cyclists.

## Figures and Tables

**Figure 1 ijerph-19-12552-f001:**
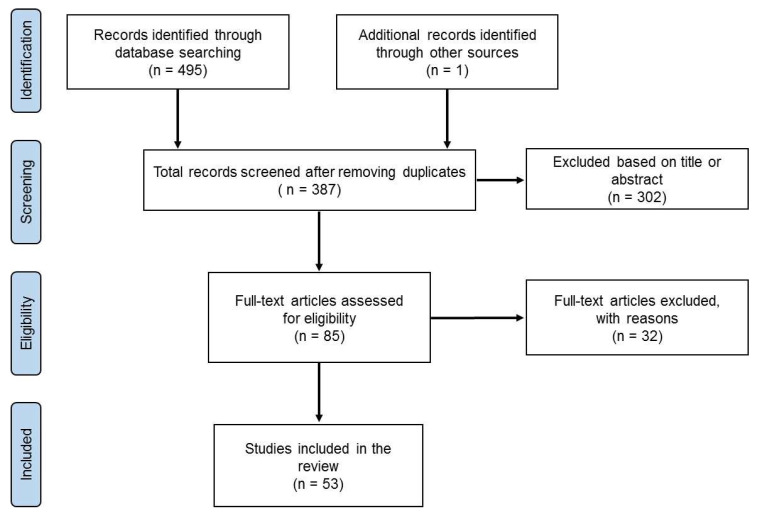
Flow diagram of search process.

**Figure 2 ijerph-19-12552-f002:**
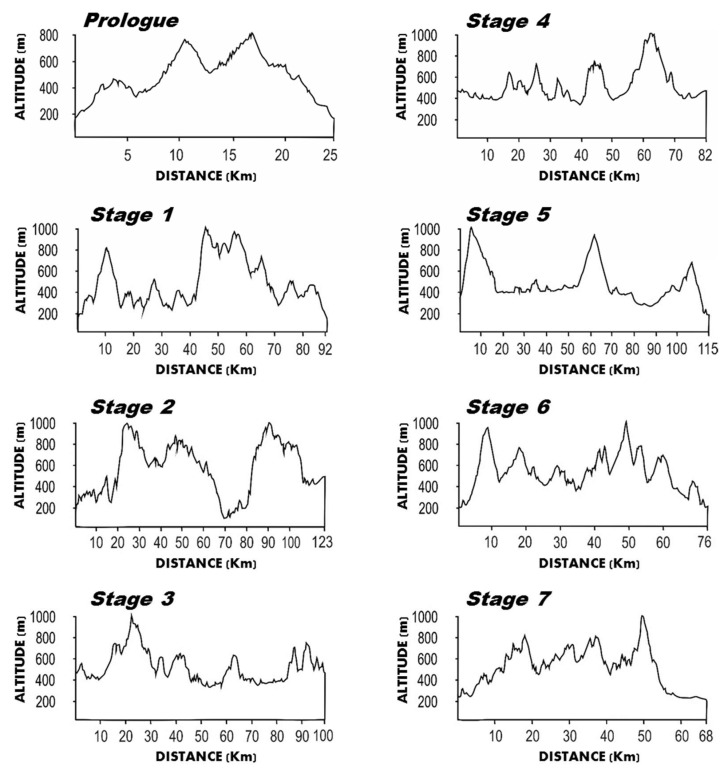
Characteristics of the Cape Epic event.

**Figure 3 ijerph-19-12552-f003:**
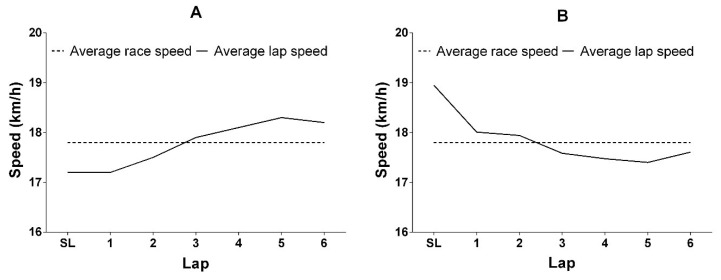
Example of a negative (**A**) and a positive (**B**) pacing profile adopted by cyclists during an XCO competition. SL: Start loop.

**Figure 4 ijerph-19-12552-f004:**
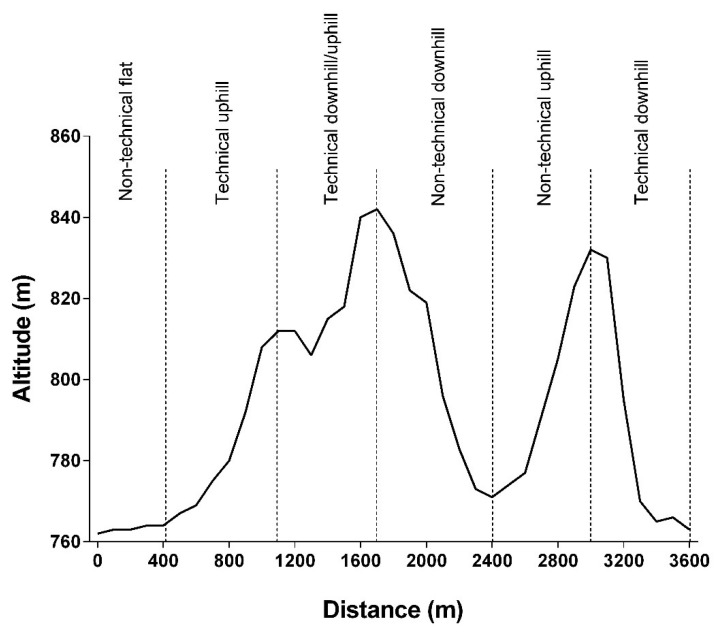
Example of an XCO circuit profile with the location of each track section for an individual lap.

**Figure 5 ijerph-19-12552-f005:**
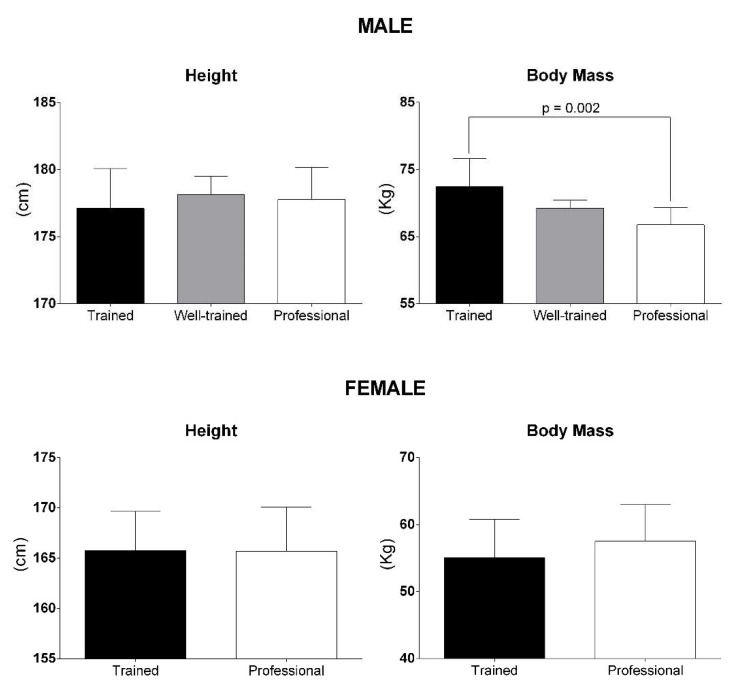
Anthropometric profile of the XC-MTB cyclists according to performance level and sex. Data are mean ± SD.

**Figure 6 ijerph-19-12552-f006:**
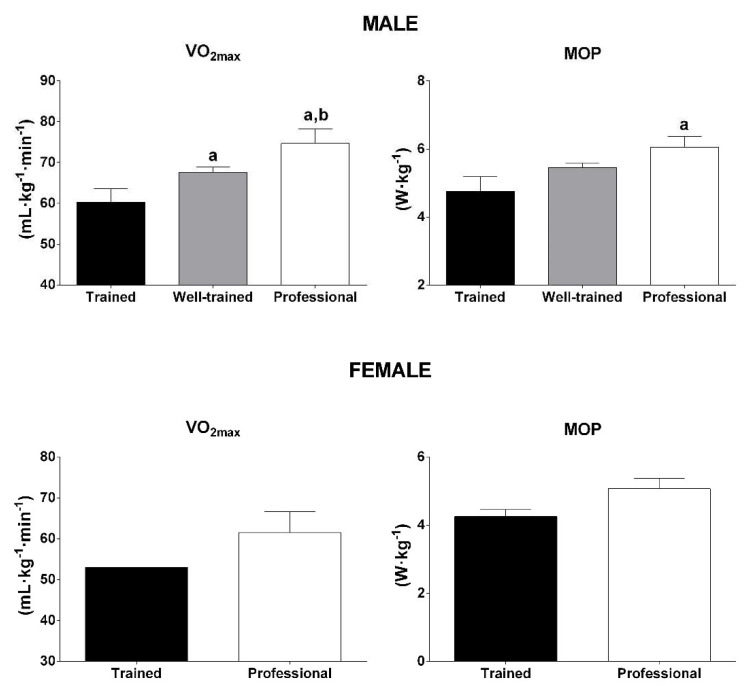
Physiological profile of the cyclists according to performance level and sex. Data are mean ± SD or only mean. One-way ANOVA test for VO_2Max_ and Kruskal–Wallis test for MOP presented *p* value: *p* < 0.01; ^a^ <0.01 vs. trained; ^b^ <0.01 vs. well-trained. VO_2Max_: maximal oxygen uptake; MOP: maximal oxidative power.

**Figure 7 ijerph-19-12552-f007:**
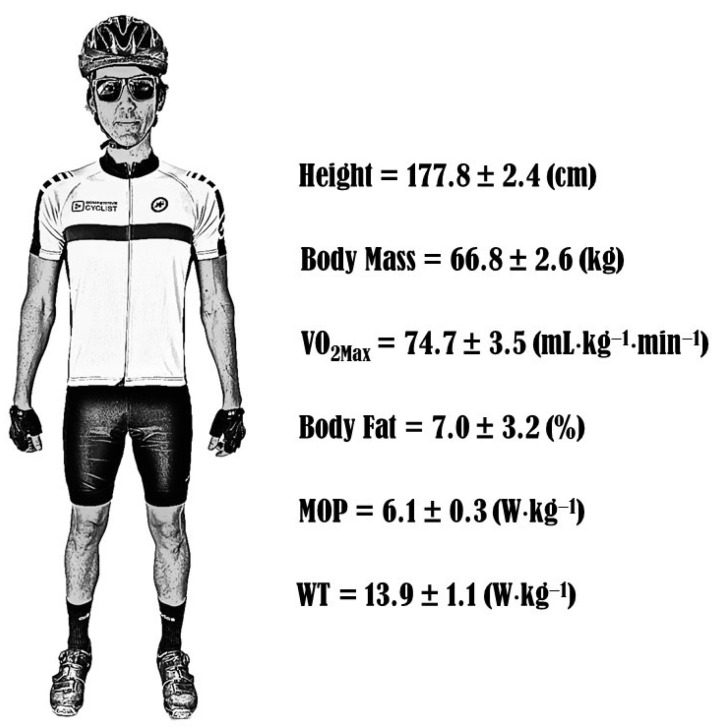
General profile of a male professional cyclist. Values are mean ± SD of the professional cyclist’s data described in Table 4 and in Section 8.1. VO_2Max_: maximal oxygen uptake; MOP: maximal oxidative power; WT: Wingate test.

**Figure 8 ijerph-19-12552-f008:**
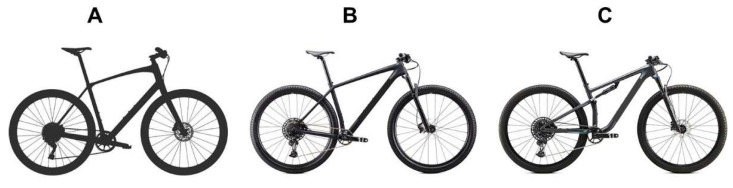
Rigid (**A**), hardtail (**B**) and full suspension (**C**) bike models for the XC-MTB events.

**Figure 9 ijerph-19-12552-f009:**
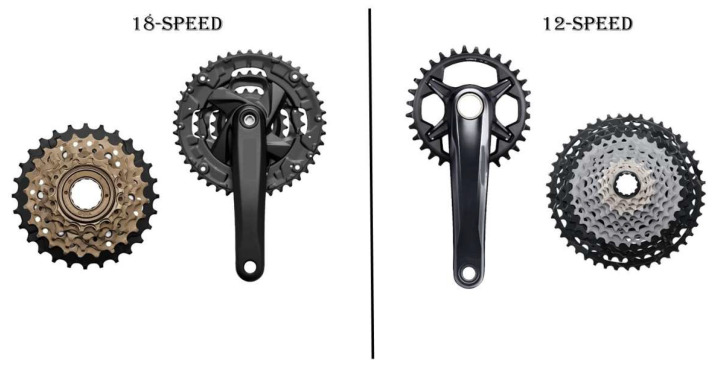
Most popular in the 1990s (18-speed) and more recent (12-speed) gear system models. Although the left system allows more speed combinations, the right one is more accurate, lighter, stronger, more efficient and easier to handle.

**Figure 10 ijerph-19-12552-f010:**
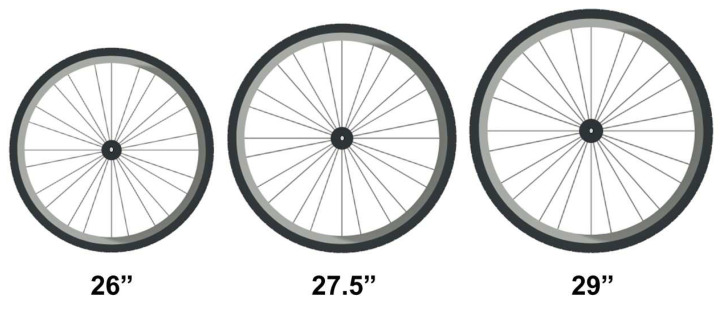
Example of the bike wheel diameters for XC throughout the years.

**Figure 11 ijerph-19-12552-f011:**
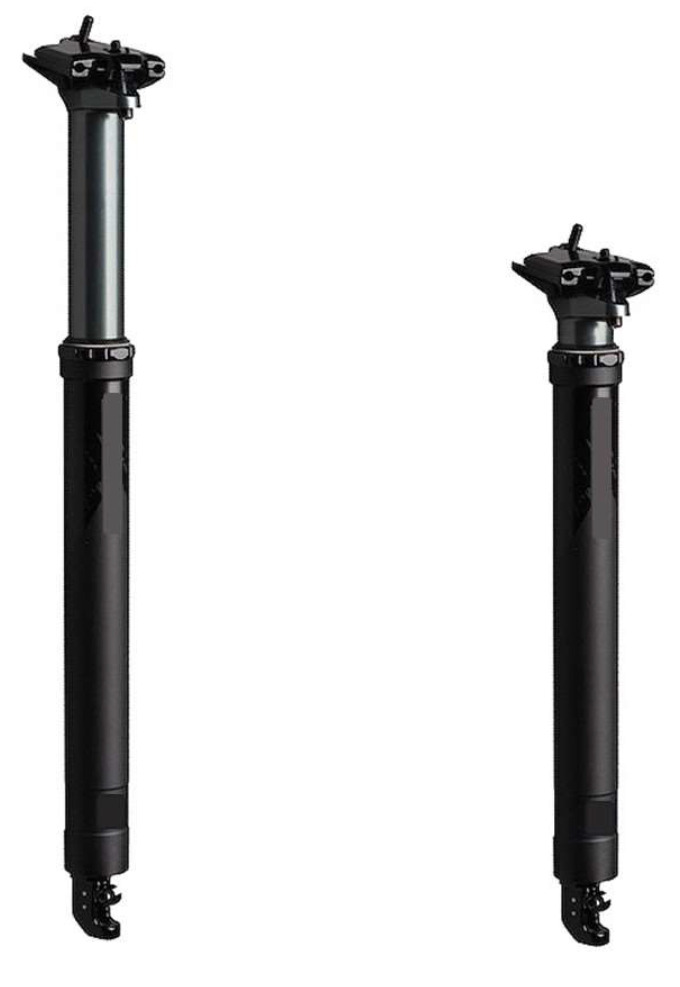
Example of a drop seatpost with original (**left**) and reduced (**right**) length/height.

**Table 1 ijerph-19-12552-t001:** Types of cross-country mountain biking events.

Event	Abbreviation	Race Time (min)	Circuit Distance (km)
Olympic cross-country	XCO	80–100	4–6
Cross-country marathon	XCM	-	20–160
Cross-country point-to-point	XCP	-	-
Cross-country short track	XCC	20–60	<2
Cross-country eliminator	XCE	< 3	0.5–1.0
Cross-country time trial	XCT	-	-
Cross-country team relay	XCR	-	-
Cross-country stage race	XCS	-	-

Data are absolute values. -: race time and/or distance are not well defined or described by UCI regulations.

**Table 2 ijerph-19-12552-t002:** Race time, physiological responses and mechanical demands to XCO competition obtained from published studies in English on the topic.

Study (Male)	Race Time (min)	HR (% HR max)	PO (W)	PO (W·kg^−1^)	PO (% PO Max)	CA (rpm)	CA-ETSNP (rpm)	Speed (km/h)
Impellizzeri et al. (2002) [3]	147 ± 15	90	-	-	-	-	-	-
Stapelfeldt et al. (2004) [9]	128 ± 17	91	246 ± 12	3.6 ± 0.2	66.9	-	-	-
Granier et al. (2018) [4]	90 ± 9	91	283 ± 22	4.3 ± 0.3	68.0	68 ± 8	83 ± 7	19.7 ± 2.1
Prinz et al. (2021) [5]	82 ± 13	91	255 ± 37	3.9 ± 0.4	68.9	64 ± 6	-	-
Study (Female)								
Stapelfeldt et al. (2004) [9]	108 ± 4	92	193 ± 1	3.1 ± 0.2	64.3	-	-	-
Prinz et al. (2021) [5]	77 ± 11	93	186 ± 18	3.6 ± 0.4	71.3	64 ± 2	-	-

Data are mean ± SD or only mean. HR: heart rate; PO: power output; CA: cadence; ETSNP: excluding the time spent not pedaling; -: not evaluated.

**Table 3 ijerph-19-12552-t003:** Percentage of time spent in different intensity zones during XCO.

Study (Method)	<10% of MOP	<FT *	Between FT and ST	>ST ^#^	>MOP
Impellizzeri et al. (2002) [3] (HR)		18 ± 10	51 ± 9	31 ± 16	
Stapelfeldt et al. (2004) [9] (PO)		39 ± 6	19 ± 6	20 ± 3	22 ± 6
Granier et al. (2018) [4] (PO)	25 ± 5	21 ± 4	13 ± 3	16 ± 3	26 ± 5
Prinz et al. (2021) [5] (PO)	28 ± 4	18 ± 8	12 ± 2	13 ± 3	30 ± 9

Data are mean ± SD. HR: heart rate; PO: power output; MOP: maximal oxidative power; FT: first threshold; ST: second threshold. <: below; >: above. *: value can be below FT or between 10% of MOP and FT; ^#^: value can be above ST or between ST and MOP.

**Table 4 ijerph-19-12552-t004:** Anthropometric and physiological profile of the cyclists according to performance level and sex.

**Male**						
**Study**	**Performance Level**	**Sample (n)**	**Height (cm)**	**BM (kg)**	**VO_2Max_ (mL·kg^−1^·min^−1^)**	**MOP** **(W·kg^−1^)**
Macrae hs-h, Hise and Allen (2000) [29]	Trained	6	179.5 ± 6.7	76.9 ± 3.6	58.4 ± 2.3	5.1 ± 0.3
Cramp et al. (2004) [30]	Trained	8	179.0 ± 6.4	69.0 ± 7.6	60.0 ± 3.7	-
Prins, Terblanche and Myburgh (2007) [31]	Trained	8	-	72.9 ± 5.6	63.6 ± 5.7	5.1 ± 0.4
Gregory, Johns and Walls (2007) [32]	Trained	11	180.2 ± 3.5	71.6 ± 6.3	64.8 ± 8.2	5.1 ± 0.4
Wirnitzer and Kornexl (2008) [15]	Trained	5	171.0 ± 4.0	63.3 ± 10.0	-	4.8 ± 0.3
Zarzeczny, Podleśny and Polak (2013) [33]	Trained	8	174.6 ± 1.1	70.3 ± 2.9	60.0 ± 1.7	-
Inoue et al. (2016) [34]	Trained	9	176.8 ± 6.7	69.6 ± 6.9	60.6 ± 4.3	4.2 ± 0.4
Hebisz et al. (2017) [35]	Trained	19	181.1 ± 9.5	73.2 ± 7.6	58.1 ± 5.8	-
Engelbrecht and Terblanche (2017) [36]	Trained	22	180.1 ± 7.9	76.4 ± 7.8	54.3 ± 7.4	4.7 ± 0.4
Costa et al. (2019) [37]	Trained	26	177.0 ± 5.0	76.0 ± 9.0	58.0 ± 7.0	-
Arriel et al. (2020) [38]	Trained	40	175.0 ± 4.0	77.8 ± 9.7	-	4.2 ± 0.7
Bazańska-Janas and Janas (2020) [39]	Trained	36	176.0 ± 17	75.8 ± 10.0	60.0 ± 6.0	5.3 ± 0.7
Inoue et al. (2021) [40]	Trained	16	175.0 ± 5.7	68.7 ± 5.6	65.4 ± 4.9	4.3 ± 0.4
Sewall and Fernhall (1995) [41]	Well-trained	10	176.7 ± 4.9	70.5 ± 8.0	68.9 ± 2.6	-
Baron (2001) [42]	Well-trained	25	179.0 ± 5.1	69.4 ± 6.5	68.4 ± 3.8	5.5 ± 0.4
Stapelfeldt et al. (2004) [9]	Well-trained	9	179.9 ± 5.9	69.4 ± 4.7	66.5 ± 2.6	5.3 ± 0.3
Inoue et al. (2012) [43]	Well-trained	10	177.9 ± 7.4	68.7 ± 7.6	68.4 ± 5.7	5.4 ± 0.5
Macdermid and Stannard (2012) [44]	Well-trained	7	176.0 ± 4.0	66.9 ± 7.7	67.6 ± 5.3	-
Smekal et al. (2015) [45]	Well-trained	24	179.0 ± 5.0	70.0 ± 4.9	64.9 ± 7.5	5.6 ± 0.6
Hebisz et al. (2020) [46]	Well-trained	20	178.4 ± 5.6	69.9 ± 9.0	67.9 ± 6.3	-
Wilber et al. (1997) [47]	Professional	10	176.0 ± 7.0	71.5 ± 7.8	70.0 ± 3.7	5.9 ± 0.3
Lee et al. (2002) [48]	Professional	7	178.0 ± 7.0	65.3 ± 6.5	78.3 ± 4.4	6.3 ± 0.5
Impellizzeri et al. (2002) [3]	Professional	5	174.6 ± 3.4	64.9 ± 4.6	75.2 ± 7.4	5.7 ±0.6
Impellizzeri et al. (2005a) [49]	Professional	13	177.0 ± 8.0	65.0 ± 6.0	72.1 ± 7.4	-
Impellizzeri et al. (2005b) [50]	Professional	12	176.0 ± 7.0	66.4 ± 5.7	76.9 ± 5.3	6.4 ± 0.6
Granier et al. (2018) [4]	Professional	8	179.0 ± 3.0	65.4 ± 3.5	79.9 ± 5.2	6.3 ± 0.4
Bejder et al. (2019) [51]	Professional	11	182.0 ± 6.0	70.2 ± 7.2	71.1 ± 7.4	-
Prinz et al. (2021) [5]	Professional	7	179.6 ± 6.7	65.3 ± 8.0	73.8 ± 2.6	5.7 ± 0.4
**Female**						
**Study**	**Performance Level**	**Sample (n)**	**Height (cm)**	**BM (kg)**	**VO_2Max_** **(mL·kg^−1^·min^−1^)**	**MOP** **(W·kg^−1^)**
Wirnitzer and Kornexl (2008) [15]	Trained	2	163.0 ± 2.1	51.0 ± 1.4	-	4.1 ± 0.6
Engelbrecht and Terblanche (2017) [36]	Trained	2	168.5 ± 4.9	59.1 ± 0.9	53.0 ± 2.8	4.4 ± 0.3
Wilber et al. (1997) [47]	Professional	10	162.0 ± 5.0	57.5 ± 4.7	57.9 ± 2.8	5.4 ± 0.4
Stapelfeldt et al. (2004) [9]	Professional	2	170.5 ± 2.1	63.0 ± 1.4	59.4 ± 1.7	4.8 ± 0.4
Prinz et al. (2021) [5]	Professional	5	164.6 ± 3.9	52.1 ± 3.1	67.3 ± 2.9	5.0 ± 0.1

Data are mean ± SD. VO_2Max_: maximal oxygen uptake; MOP: maximal oxidative power; BM: body mass; -: not evaluated. Note from the authors: manuscripts that did not report anthropometric data were not included in the table.

## Data Availability

Data are contained within the manuscript.

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
