# Peer review of "Current Perspectives of Cross-Country Mountain Biking: Physiological and Mechanical Aspects, Evolution of Bikes, Accidents and Injuries"

_ijerph, 2022, doi:10.3390/ijerph191912552_

Round 1

Reviewer 1 Report

I consider that the manuscript fits in the special issue addressed.

The present review is very interesting and presents relevant data for readers, researchers and trainers of mountain biking

There are a lot of consistent data regarding the topic and, before publication I would like to suggest some modifications

-        Please review the language; the are some typos along the text

-        Add one or two more columns in the table 1 regarding the characteristics of races

-        Are there other physiological responses besides HR and Power that could be described in the manuscript?

-        Although the authors have used a large board of references, I am not convinced that some studies of Impellizzeri and cols. Has some mistakes, mainly regarding mechanical responses of MTB (carefully check those studies in details!)

-        Add one more sentencer about figure 1; I consider that if you add a fig on your manuscript, you should explore it.

-        Increase fonts in some fig, e.g., fig 2

-        Figure 3 refers to what type of XC? please describe in the text, and in the figure and explore it further in the text

-        IN the topic 7.2, I consider important to add data about what is the minimum requirement to compete in the elite of XC-MTB

-        7.2.2: I do not like the term “anaerobic”. In which metabolic pathways does oxygen not participate? perhaps fast glycolysis is more suitable (see the texts of Brooks and Robergs)

-        7.3: is it possible to cover intervals on the data from fig 6; maybe Sem or SD or CI

-        Maybe connect topics 7.3 and 8 better, since the athlete's fit on the bike is also related to anthropometer, bike size. In addition, on item 8, comment the fact that athletes often sweat bikes smaller than the recommended fit size, trying to carry less weight and be more agile but this is a it is a mistaken fact, as often the weight difference between an S and an XL bike is less than 200-250 grams

-        8.1 describe the lock system of suspension and its applicability

-        9. despite few or no studies on accidents and MTB, add your opinion on what can or is done to minimize and/or prevent accidents in the race and how athletes can train for this

Author Response

Dear Editor and Reviewers,

We would like to thank for the insightful comments regarding our manuscript. All recommended modifications were provided as you can see below. We did our better and we appreciate your time and effort in reviewing our paper. Thanks.

REVIEWER 1:

Comments:

I consider that the manuscript fits in the special issue addressed.

The present review is very interesting and presents relevant data for readers, researchers and trainers of mountain biking

There are a lot of consistent data regarding the topic and, before publication I would like to suggest some modifications.

- Please review the language; the are some typos along the text.

R.: The manuscript was read by a native English-speaking colleague.

- Add one or two more columns in the table 1 regarding the characteristics of races.

R.: We added race time and circuit distance. Please, note that these items are not well defined in some events.

- Are there other physiological responses besides HR and Power that could be described in the manuscript?

R.: Thank you for your suggestion. Unfortunately, there is a limited number of studies and analyses on the topic. We found a study that analyzed oxygen consumption (DOI: 10.3389/fphys.2018.01062). However, the simulated and XCO race performed in the study did not comply with UCI regulations, such as race time. Thus, we do not report this data in order to avoid bias.

- Although the authors have used a large board of references, I am not convinced that some studies of Impellizzeri and cols. Has some mistakes, mainly regarding mechanical responses of MTB (carefully check those studies in details!).

R.: Thank you for your attention. We carefully checked the data of the studies.

- Add one more sentencer about figure 1; I consider that if you add a fig on your manuscript, you should explore it.

R.: We added one more sentence about figure 1.

- Increase fonts in some fig, e.g., fig 2.

R.: We have increased fonts of the fig 2, 4 and 5.

- Figure 3 refers to what type of XC? please describe in the text, and in the figure and explore it further in the text.

R.: Figure 3 refers to XCO event. We have described more details about figure 3 in the text.

- IN the topic 7.2, I consider important to add data about what is the minimum requirement to compete in the elite of XC-MTB.

R.: These data were described in the topic 7.3 and figure 6. We clarified this information in the topic 7.3.  

- 7.2.2: I do not like the term “anaerobic”. In which metabolic pathways does oxygen not participate? perhaps fast glycolysis is more suitable (see the texts of Brooks and Robergs).

R.: We changed the terms along the text.

- 7.3: is it possible to cover intervals on the data from fig 6; maybe Sem or SD or CI.

R.: Figure 6 data were changed to mean ± SD.

- Maybe connect topics 7.3 and 8 better, since the athlete's fit on the bike is also related to anthropometer, bike size. In addition, on item 8, comment the fact that athletes often sweat bikes smaller than the recommended fit size, trying to carry less weight and be more agile but this is a it is a mistaken fact, as often the weight difference between an S and an XL bike is less than 200-250 grams.

R.: Although we have left the above topics separate, we have made considerable changes to the second. On end of topic 8.2, we have added information about the fact of using smaller bikes was added, in order to improve agility and at the same time the possibility of a bike with greater distance between axles to present greater stability in ups and/or descents. We have also extended the part about bike fit.

- 8.1 describe the lock system of suspension and its applicability.

R.: We added more information about lock system in the text.

- 9. despite few or no studies on accidents and MTB, add your opinion on what can or is done to minimize and/or prevent accidents in the race and how athletes can train for this.

R.: We added more information about prevent accidents measures in the race, suggesting possible strategies for this.

Reviewer 2 Report

First of all, I would like to congratulate the authors for big effort to do this work.

Regarding the manuscript, this presents a very interesting and novel topic, although as written, it seems more like a general sports newspaper article than a research work.

Since there are many related articles, it is suggested that authors can do a systematic reviews to get more powerful conclusions. To do that I suggest using some specific and well known methodology such a PRISMA, describing the methodological process according to PRISMA guideline. As a benefit, this method would permit to find some missing items in previous version of the document.

Other comments

Line 137: I don't agree with these specific cadences in laboratories, due to the fact that, first, it is not true that it is stated that 60rpm is more economical in trained subjects, which is full of references supporting it. To support this, it is obvious that none of the professional cyclists use generally such a low cadence in competition.

Line 140: I would suggest including the word "high" in this sentence to better understand (requiring a highly CA variation…".

Lines 152 to 155: There is a huge controversy nowadays regarding this terminology to identify both physiological events. I would suggest authors not to use this words and look for other identification for these events (i.e. first and second ventilatory or lactic thresholds...)

Line 178: I miss some kind of reference which support this statement

Line 204: I wonder if these thresholds were established previously in laboratory. It might help to understand this sentence

Line 216: I guess authors mean "between" instead of “below”.

Line 232: I would highlight that the prologue is remarkably shorter than the others

Lines 333 to 340: I think authors suggest the opposite in the first and in the last part of the paragraph. I suggest authors to rephrase it.

Lines 341 to 343: This is so obvious that in my opinion it is not necessary to include in the manuscript.

Lines 448 to 450: I suggest authors to include any additional reference supporting differences between carbon or alloy frames.

Line 492: I suggest authors to include any comment regarding how FS has better grip in steep-unpaved climbs, and of course if possible, some reference.

Line 506: I guess authors mean "soil"

Line 515: I suggest including any photo of a seat-post.

Line 531: Please, consider to introduce another issue: "it allows easing the transmission system reducing friction

Line 574: I suggest authors to introduce here the term "boost hubs", which could explain inefficiencies in the firsts models of 29" wheels. Boost wheels have a wider axle, which means that the spacing between your hub’s flanges can be increased. By increasing the width of the hub flanges the bracing angles of the spokes in the wheel are improved, getting a stiffer and ultimately more efficient wheel

Line 583: Please, consider introducing the fact that this kind of seatposts are heavier than rigid-carbon-ones, as another reason for not been used for some cyclists.

Author Response

Dear Editor and Reviewers,

We would like to thank for the insightful comments regarding our manuscript. All recommended modifications were provided as you can see below. We did our better and we appreciate your time and effort in reviewing our paper. Thanks.

REVIEWER 2:

Comments:

            First of all, I would like to congratulate the authors for big effort to do this work.

            Regarding the manuscript, this presents a very interesting and novel topic, although as written, it seems more like a general sports newspaper article than a research work.

            Since there are many related articles, it is suggested that authors can do a systematic reviews to get more powerful conclusions. To do that I suggest using some specific and well known methodology such a PRISMA, describing the methodological process according to PRISMA guideline. As a benefit, this method would permit to find some missing items in previous version of the document.

R.: We carried out a systematic search according to your suggestion. We have added the respective manuscripts on specific sections, according to the reported parameters (i.e., on table 4 about anthropometric data we did not included studies without these parameters).

Other comments

Line 137: I don't agree with these specific cadences in laboratories, due to the fact that, first, it is not true that it is stated that 60rpm is more economical in trained subjects, which is full of references supporting it. To support this, it is obvious that none of the professional cyclists use generally such a low cadence in competition.

R.: We have rephrased the sentence, added more reference reporting that the ability to identify an ‘‘optimal’’ cadence is limited [DOI: 10.1080/17461390802684325].

Line 140: I would suggest including the word "high" in this sentence to better understand (requiring a highly CA variation…".

R.: We added the word “highly” accordingly.

Lines 152 to 155: There is a huge controversy nowadays regarding this terminology to identify both physiological events. I would suggest authors not to use this words and look for other identification for these events (i.e. first and second ventilatory or lactic thresholds...)

R.: Thank you for your suggestion. We have changed the terms to “first and second thresholds” along the text.

Line 178: I miss some kind of reference which support this statement

R.: We have modified and cited a study on this line.

Line 204: I wonder if these thresholds were established previously in laboratory. It might help to understand this sentence

R.: The thresholds were established previously in laboratory. We have added this information in the sentence.

Line 216: I guess authors mean "between" instead of “below”.

R.: Sorry for the confusing sentence. In this, the word is “below”. The values 55% and 68% correspond to PO and HR, respectively. We rewrite the sentence.

Line 232: I would highlight that the prologue is remarkably shorter than the others

R.: We agree with this affirmation and we have highlighted on the text. (lines 247-249)

Lines 333 to 340: I think authors suggest the opposite in the first and in the last part of the paragraph. I suggest authors to rephrase it.

R.: Thank you for your suggestion. We have rephrased the paragraph.

Lines 341 to 343: This is so obvious that in my opinion it is not necessary to include in the manuscript.

R.: We agreed and deleted the sentence in the manuscript.

Lines 448 to 450: I suggest authors to include any additional reference supporting differences between carbon or alloy frames.

R.: We have modified this part accordingly

Line 492: I suggest authors to include any comment regarding how FS has better grip in steep-unpaved climbs, and of course if possible, some reference.

R.: Thank you for your suggestion. We added this information in the text.

Line 506: I guess authors mean "soil"

R.: We have corrected accordingly

Line 515: I suggest including any photo of a seat-post.

R.: We have added a photo of a seatpost.

Line 531: Please, consider to introduce another issue: "it allows easing the transmission system reducing friction

R.: Thank you for your suggestion. We added this issue in the sentence.

Line 574: I suggest authors to introduce here the term "boost hubs", which could explain inefficiencies in the firsts models of 29" wheels. Boost wheels have a wider axle, which means that the spacing between your hub’s flanges can be increased. By increasing the width of the hub flanges the bracing angles of the spokes in the wheel are improved, getting a stiffer and ultimately more efficient wheel

R.: Thank you for this suggestion. We have added the above information to the text.

Line 583: Please, consider introducing the fact that this kind of seatposts are heavier than rigid-carbon-ones, as another reason for not been used for some cyclists

R.: We added your suggestion in the sentence.
